# Convergence Rates for Hestenes' Gram–Schmidt Conjugate Direction Method without Derivatives in Numerical Optimization

**Ivie Stein, Jr. [1] and Md Nurul Raihen [2,*]**

1   Department of Mathematics and Statistics, The University of Toledo, Toledo, OH 43606, USA
2   Department of Mathematics and Statistics, Stephen F. Austin State University, Nacogdoches, TX 75962, USA
*   Correspondence: nurul.raihen@sfasu.edu; Tel.: +1-313-378-7353

**Abstract:** In this work, we studied convergence rates using quotient convergence factors and root convergence factors, as described by Ortega and Rheinboldt, for Hestenes' Gram–Schmidt conjugate direction method without derivatives. We performed computations in order to make a comparison between this conjugate direction method, for minimizing a nonquadratic function $f$, and Newton's method, for solving $\nabla f = 0$. Our primary purpose was to implement Hestenes' CGS method with no derivatives and determine convergence rates.

**Keywords:** conjugate direction method; Gram–Schmidt conjugate process without derivatives; convergence rates; numerical optimization

## 1. Introduction

The conjugate gradient (CG) and conjugate direction (CD) methods have been extended to the optimization of nonquadratic functions by several authors. Fletcher and Reeves [1] gave a direct extension of the conjugate gradient (CG) method. An approach to conjugate direction (CD) methods using only function values was developed by Powell [2]. Davidon [3] developed a variable metric algorithm, which was later modified by Fletcher and Powell [4]. According to Davidon [3], variable metric methods are considered to be very effective techniques for optimizing a nonquadratic function.

In 1952, Hestenes and Stiefel [5] developed conjugate direction (CD) methods for minimizing a quadratic function defined on a finite dimensional space. One of their objectives was to find efficient computational methods for solving a large system of linear equations. In 1964, Fletcher and Reeves [1] extended the conjugate gradient (CG) method of Hestenes and Stiefel [5] to nonquadratic functions. The method presented here is related to those described by G.S. Smith [6], M.J.B. Powell [2] and W.I. Zangwill [7]. The method of Smith is also described by Fletcher [8] on pp. 9–10, Brent [9] on p. 124 and Hestenes [10] on p. 210. In addition to that, Nocedal [11] explored the possibility of nonlinear conjugate gradient methods converging without restarts and with the use of practical line search. In the field of numerical optimization, a number of additional authors, including Kelley [12], Zang and Li [13], among others, investigated a wide range of approaches in the use of conjugate direction methods.

The primary purpose of this work is to implement Hestenes' Gram–Schmidt conjugate direction method without derivatives, which uses function values with no line searches. We will refer to this method as the GSCD method; Hestenes refers to it as the CGS method. We illustrate the procedure numerically, computing asymptotic constants and the quotient convergent factors of Ortega and Rheinboldt [14]. In reference to Hestenes [10], p. 202, where he states that the CGS has Newton's algorithm as its limit, Russak [15] shows that n-step superlinear convergence is possible. We verify numerically that the GSCD procedure converges quadratically under appropriate conditions.

As for notation, we use capital letters, such $A, B, C, \ldots$, to denote matrices and lower case letters, such as $a, b, c, \ldots$, for scalars. The value $A^*$ denotes the transpose of matrix $A$. If $F$ is a real-valued differentiable function of $n$ real variables, we denote its gradient at $x$ by $F'(x)$ and the Hessian of $F$ at $x$ by $F''(x)$. We use subscripts to distinguish vectors and superscripts to denote components when these distinctions are made together, for example, $x_k = (x_k^1, \ldots, x_k^n)$.

The method of steepest descent is due to Cauchy [16]. It is one of the oldest and most obvious ways to find a minimum point of a function $f$.

There are two versions of steepest descent. The one due to Cauchy, which we call an iterative method, uses line searches and another, described by Eells [17] in Equation (10), p. 783, uses a differential equation of steepest descent. In Equation (4.3) we describe another version of the differential equation of steepest descent. However, numerically, both have flaws. The iterative method is generally quite slow, as shown by Rosenbrock [18] in his banana valley function.

Newton's method applied to $\nabla f = 0$, where $f$ is a function to be minimized, is another approach for finding a minimum of the function $f$. Newton's method has rapid convergence, but it is costly because of derivative evaluations. Hestenes' CGS method without derivatives [10], p. 202, has Newton's method as its limit as $\sigma \to 0$.

If the minimizing function is strongly convex quadratic and the line search is exact, then, in theory, all choices for the search direction in standard conjugate gradient algorithms are equivalent. However, for nonquadratic functions, each choice of the search direction leads to standard conjugate gradient algorithms with very different performances [19].

In this article, we investigate quotient convergence factors and root convergence factors. We computationally compare the conjugate Gram–Schmidt direction method with Newton's method. There are other types of convergence for the conjugate gradient, the conjugate direction, the gradient method, Newton's method and the steepest descent method, such as superlinear convergence [20–22], Wall [23] root convergence and Ostrowski convergence factors [24], but, for the sake of this research, quotient convergence is the one that is the most appropriate for the quadratic convergence.

In this article, the well-known conjugate directions algorithm, for minimizing a quadratic function, is modified to become an algorithm for minimizing a nonquadratic function, in the manner described in Section 2. The algorithm uses the gradient estimates and Hessian matrix estimates described in Section 3. In Section 4, a test example for minimizing a nonquadratic function by the developed conjugate direction algorithm without derivatives is analyzed. The advantage of this approach compared to Newton's method is efficiency. The proposed approach is justified in sufficient detail. The results obtained are of certain theoretical and practical interest for specialists in the theory and methods of optimization.

## 2. Methodology

In this section, we present a class of CD algorithms for minimizing functions defined on an $n$-dimensional space. The reader is directed to refer to Stein [25] and Hestenes [10], pp. 135–137 and pp. 199–202, respectively, for more details.

### 2.1. The Method of CD

Let $A$ be a positive definite real symmetric $n \times n$ matrix, let $k$ be a constant $n$-vector and let $c$ be a fixed real scalar. Throughout this section, $F$ denotes the function defined on Euclidean $n$-space $E_n$ by

$$F(x) = \frac{1}{2} x^* A x - k^* x + c, \tag{1}$$

where $x$ is in $E_n$.

Suppose $1 \leq m \leq n$. Let $S_m$ be the linear subspace spanned by the set $\{p_1, \ldots, p_n\}$ of $m$ linearly independent and, hence, nonzero vectors. Let $x_1$ be any vector in $E_n$. Then, the $m$-dimensional plane $P_m$ through $x_1$ obtained by translating the subspace $S_m$ is defined by

$$P_m = \left\{ x : x = x_1 + \alpha_1 p_1 + \ldots + \alpha_m p_m, \ \alpha_i \in \mathbb{R} \ (i = 1, \ldots, m) \right\}. \tag{2}$$

Two vectors, $p$ and $q$, in $E_n$ are said to be $A$-orthogonal or conjugate if $p^* A p = 0$. A set $\{p_1, \ldots, p_m\}$ of nonzero vectors in $E_n$ is said to be mutually $A$-orthogonal or mutually conjugate if

$$p_i^* A p_j = 0 \quad for \quad i \neq j \ (i = 1, \ldots, m).$$

**Theorem 1** ([25]). *Let $S_m$ be a subspace of $E_n$, where $\{p_1, \ldots, p_n\}$ is a basis for $S_m$, $1 \leq m \leq n$. Further assume that $p_1, \ldots, p_m$ is a mutually A-orthogonal set of vectors. Let $x_1$ be any vector in $E_n$. Let $x$ be in $P_m$. Then, the following conditions are equivalent:*

1. *$x$ minimizes F on $P_m$.*
2. *$F'(x)$, the gradient of F at $x$, is orthogonal to the subspace $S_m$.*
3. *$x = x_1 + a_1 p_1 + \ldots + a_m p_m$, where $a_i = \dfrac{c_i}{d_i}$, $c_i = -p_i^* F'(x_1)$, $d_i = p_i^* A p_i$, $i = 1, \ldots, m$.*

*Let $x_i = x_1 + a_1 p_1 + \ldots + a_i p_i$, $i = 1, \ldots, m$. Then the quantity $c_i$ defined in (3) above is also given by*

$$c_i = -p_i^* F'(x_i), \quad i = 1, \ldots, m.$$

*Then, there is a unique vector $x_0$ in the m-dimensional plane $P_m$ through $x_1$ translating $S_m$ such that $x_0$ minimizes the function F given by (1) on $P_m$.*

**Proof.** First, we are going to show that $F$ has at least one minimizing vector in $P_m$. Let $p$ be any vector in $P_m$ and let $M = F(p)$. Since $A$ is positive definite, there is an $R \in \mathbb{R} > 0$ such that $||x|| > R$ implies $F(x) > M$. Hence, $F(x) \leq M$ implies $||x|| \leq R$. Since $C = \{x : ||x|| \leq R\} \cap P_m$ is a compact set in $E_n$ on which $F$ assumes values and is continuous, then $F$ has at least one minimizing vector $p_0$ in the compact set $C$. Outside this compact set $C$, $F$ assumes only larger values. Thus, $p_0$ is a minimizing vector for $F$ in $P_m$.

To show that (1) implies (2), assume that $x$ minimizes $F$ on $P_m$. Then,

$$p_j^* F'(x) = \frac{dF}{d\alpha}(x + \alpha p_j)\Big|_{\alpha=0} = 0, \tag{3}$$

for $j = 1, \ldots, m$. Thus,

$$p_j^* F'(x) = 0 \quad (j = 1, \ldots, m). \tag{4}$$

So $F'(x)$ is orthogonal to every vector in the basis of $S_m$ and, hence, is orthogonal to $S_m$.

To show (2) implies (1), suppose that $F'(x)$ is orthogonal to $S_m$. Let $v$ be any vector in $P_m$. We are going to show that $F(v) > F(x)$ unless $v = x$. By Taylor's theorem we have the following:

$$F(v) = F(x) + (v - x)^* F'(x) + \frac{1}{2}(v - x)^* A(v - x). \tag{5}$$

Since $(v - x)$ is a vector in $S_m$, then $(v - x)^* F'(x) = 0$. In addition, $(v - x)^* A(v - x) > 0$ unless $v = x$, because A is positive definite. Thus,

$$F(v) > F(x) \qquad \text{unless} \quad v = x. \tag{6}$$

Hence, $x$ is a unique absolute minimum for $F$ in $P_m$.

Now we can prove that there is a unique vector $x_0$ in $P_m$ minimizing $F$ on $P_m$. Earlier we established that there is at least one minimizing vector $p_0$ for $F$ in $P_m$. Since (1) implies (2), then $F'(p_0)$ is orthogonal to $S_m$. From the proof of (2) implies (1), it now follows that $p_0$ is a unique absolute minimum for $F$ in $P_m$.

To show that (2) implies (3), let $x = x_1 + a_1 p_1 + \ldots + a_m p_m$ since $x$ is in $P_m$, and assume that $F'(x)$ is orthogonal to the subspace $S_m$. We are going to show that $a_i = \dfrac{c_i}{d_i}$, where $c_i = -p_i^* F'(x_1)$, $d_i = p_i^* A p_i$, $i = 1, \ldots, m$. Note that $Ax = Ax_1 + a_1 A p_1 + \ldots + a_m A p_m$. In addition, $Ax - k = Ax_1 - k + a_1 A p_1 + \ldots + a_m A p_m$.

Since $F'(x) = Ax - k$, then

$$F'(x) = F'(x_1) + a_1 A p_1 + \ldots + a_m A p_m.$$

For $i = 1, \ldots, m$, we have

$$p_i^* F'(x) = p_i^* F'(x_1) + a_1 p_i^* A p_1 + \ldots + a_m p_i^* A p_m.$$

Since $\{p_1, \ldots, p_m\}$ is a mutually A-orthogonal set of vectors, then $p_i^* F'(x) = p_i^* F'(x_1) + a_i p_i^* A p_i$, $i = 1, \ldots, m$. Since $F'(x)$ is orthogonal to the subspace $S_m$, then $p_i^* F'(x) = 0$, $i = 1, \ldots, m$. Thus, $a_i p_i^* A p_i = -p_i^* F'(x_1)$. Since $p_i \neq 0$, $i = 1, \ldots, m$, and $A$ is positive definite, then $p_i^* A p_i \neq 0, i = 1, \ldots, m$. If we let $c_i = -p_i^* F'(x_1)$ and $d_i = p_i^* A p_i$, then

$$a_i = \frac{c_i}{d_i}, \qquad i = 1, \ldots, m. \tag{7}$$

To show that (3) implies (2), we can use what was established in the previous proof. An indication of this is proved below.

Suppose that $x = x_1 + a_1 p_1 + \ldots + a_m p_m$, where $a_i = \dfrac{c_i}{d_I}$, $c_i = -p_i^* F'(x_1)$, $d_i = p_i^* A p_i$, $i = 1, \ldots, m$. We want to show that $p_i^* F'(x) = 0, i = 1, \ldots, m$. Since

$$p_i^* F'(x) = p_i^* F'(x_1) + a_i p_i^* A p_i,$$

and $a_i = \dfrac{-p_i^* F(x_1)}{p_i^* A p_i}$, then we have $p_i^* F'(x) = 0, i = 1, \ldots, m$. Hence, $F'(x)$ is orthogonal to $S_m$. Thus, (1)–(3) are equivalent.

Now we are going to show that the quantity $c_i$ defined by $c_i = -p_i^* F'(x_1)$, $i = 1, \ldots, m$, in (3) is also given by $c_i = -p_i^* F'(x_i)$, $i = 1, \ldots, m$.

Since $x_{i+1} = x_i + a_i p_i$, $i = 1, \ldots, (m-1)$, then

$$Ax_{i+1} = Ax_i + a_i A p_i,$$
$$Ax_{i+1} - k = Ax_i - k + a_i A p_i,$$
$$F'(x_{i+1}) = F'(x_i) + a_i A p_i \quad i = 1, \ldots, (m-1).$$

Thus,

$$F'(x_{i+1}) = F'(x_1) + a_i A p_i + \ldots + a_1 A p_1, \quad i = 1, \ldots, (m-1), \tag{8}$$

and, by conjugacy of $\{p_1, \ldots, p_m\}$, we have

$$p_i^* F'(x_i) = p_i^* [F'(x_1) + a_1 A p_1 + \ldots + a_{i-1} A p_{i-1} = p_i^* F'(x_1), \quad i = 1, \ldots, m. \tag{9}$$

Hence,

$$p_i^* F'(x_1) = p_i^* F'(x_i), \quad i = 1, \ldots, m. \tag{10}$$

This completes the proof of the theorem. $\square$

### 2.2. A Class of Minimization Algorithms

Now, we shall describe a class of minimization algorithms known as the method of CDs. The significance of the formulas given in (3) of Theorem 1 is indicated below.

Suppose $\{p_1, \ldots, p_m\}, 1 \leq m \leq n$, is a conjugate set of nonzero vectors and that $P_m$ is the $m$-dimensional plane through $x_1$ obtained by translating the subspace $S_m$ generated by $\{p_1, \ldots, p_m\}$. Then, the minimum of $F$ given by (1) on $P_m$ is attained at $x_0$, which we will call $x_{m+1}$, where $x_{m+1} = x_1 + a_1 p_1 + \ldots + a_m p_m$, refer to Theorem 1. Now we assume that $p_{m+1}$ is a nonzero vector that has been constructed to be conjugate to $p_i, i = 1, \ldots, m$, and let $P_{m+1}$ denote the $(m+1)$-dimensional plane through $x_1$ obtained by translating the subspace $S_{m+1}$ generated by $\{p_1, \ldots, p_m, p_{m+1}\}$. It turns out that it is not necessary to solve

a new $(m+1)$-dimensional minimization problem to determine the minimizing vector $x_{m+2}$ on $P_{m+1}$.

The minimizing vector $x_{m+2}$ on $P_{m+1}$ is obtained by a one-dimensional minimization of $F$ about the vector $x_{m+1}$ in the direction $p_{m+1}$. This follows directly from the following formulas found in Theorem 1:

$$x_{m+2} = x_{m+1} + a_{m+1} p_{m+1},$$

and

$$a_{m+1} = \frac{c_{m+1}}{d_{m+1}}, \qquad c_{m+1} = -p_{m+1}^* F'(x_{m+1}), \qquad d_{m+1} = p_{m+1}^* A p_{m+1}.$$

Note that $a_{m+1}$ depends upon $x_{m+1}$ and $p_{m+1}$ and explicitly involves no other $x$ or $p$ terms. Thus, the minimizing vector $x_{m+1}$ on $P_m$ results from $m$ consecutive one-dimensional minimizations starting at $x_1$ and preceding along the CDs $p_1, \ldots, p_m$ successively. The ways of obtaining a mutually conjugate set $\{p_1, \ldots, p_m\}$ of vectors are not specified in general. Thus, the method of CDs is really a class of algorithms, where a specific algorithm depends upon the choice of $\{p_1, \ldots, p_m\}$. In practice, the vector $p_k$, $k = 1, \ldots, m$, needed for the $(k+1)^{th}$ iteration in finding $x_{k+1}$, $k = 1, \ldots, m$, is usually constructed from information obtained at the $k^{th}$ iteration, $k = 1, \ldots, m$. The following class of algorithms is referred to as the method of CDs: for $k = 1, \ldots, n$, we find

$$x_{k+1} = x_k + a_k p_k,$$

$$a_k = \frac{c_k}{d_k}, \qquad c_k = -p_k^* F'(x_1), \qquad d_k = p_k^* A p_k.$$

Alternatively, $c_k$ may be given by

$$c_k = -p_k^* F'(x_k).$$

If $F'(x_m) = 0$ for $1 \le m \le n$, then the algorithm terminates and $x_m$ minimizes $F$ on $E_n$. Furthermore, any algorithm terminates in $n$ steps or less since $F$ is quadratic.

*2.3. Special Inner Product and the Gram–Schmidt Process*

Let $A$ be a positive definite symmetric $n \times n$ matrix. Define a special inner product $(x, y)$ by

$$(x, y) = x^* A y,$$

where $x$ and $y$ are column vectors.

Let

$$u_1^* = (1, 0, \cdots, 0), \quad u_2^* = (0, 1, 0, \cdots, 0) \quad \text{and} \quad u_n^* = (0, 0, \cdots, 0, 1).$$

Then, using the special inner product above, we apply the Gram–Schmidt process to the linearly independent vectors $u_1, u_2, \ldots, u_n$ to obtain a set of mutually $A$-orthogonal vectors $p_1, p_2, \ldots, p_n$, where the property of $A$-orthogonality is relative to the special inner product as performed by Hestenes and Stieffel [5] on p. 425.

## 3. Results

A brief description of the CG method is given below using a quadratic function:

$$F(x) = \frac{1}{2} x^* A x - k^* x + c.$$

The CG method is the CD method, which is described previously, with the first CD being in the direction of the negative gradient of function $F$. The remaining CDs can be determined in a variety of ways, and the CG procedure described by Hestenes [10] is given below.

*3.1. CG—Algorithms for Nonquadratic Approximations*

One can apply the CG method to the quadratic function in $z$, namely $F(z)$, to obtain a minimum of $F(z)$. Let $f$ be a function of $n$ variables, then

$$F(z) = f(x_1) + (f'(x_1))^* z + \frac{1}{2} z^* f''(x_1) z.$$

Assume that a Hessian matrix is a positive definite symmetric matrix, which implies that $F(z)$ has a unique minimum $\bar{z}_{\min}$. Then,

$$\nabla F(z) = f'(x_1) + f''(x_1) z.$$

Applying Newton's method to $\nabla F(z) = 0$, we get

$$f'(x_1) + f''(x_1) z = 0,$$
$$(f''(x_1))^{-1}(f'(x_1)) + z = 0 \quad \text{multiplied by} \quad (f''(x_1))^{-1},$$
$$\bar{z}_{\min} = -(f''(x_1))^{-1}(f'(x_1)).$$

**Remark 1.** *We solved* $\nabla F(\bar{z}) = \bar{0}$ *directly to obtain* $\min F(z)$.

In general, Newton's method is used to solve $\vec{f}(\bar{z}) = \bar{0}$ for $\bar{z}$. It is given by

$$z_{n+1} = z_n - J_n^{-1} f(z_n), \qquad n = 0, 1, 2, \ldots$$

where $z_0$ is an initial guess and $J_n$ is the Jacobian matrix, i.e.,

$$J_n = \begin{pmatrix} \frac{\partial f^{(1)}(z_n)}{\partial z^1} & \cdots & \frac{\partial f^{(1)}(z_n)}{\partial z^n} \\ \vdots & \ddots & \vdots \\ \frac{\partial f^{(1)}(z_n)}{\partial z^1} & \cdots & \frac{\partial f^{(1)}(z_n)}{\partial z^n} \end{pmatrix}.$$

Now, we apply Newton's method by taking $\bar{f}$ to $\nabla F$ and assuming that $F$ and its second partial derivatives are continuous. So, one can apply Newton's method to $\nabla F(z) = \bar{0}$, with $z_1 = 0$ as the initial point, to obtain the minimum point $\bar{z}_{min}$ of $F$, where

$$z_{n+1} = z_n - J_n^{-1}(\nabla F z_n) = z_n - (F''(x_1))^{-1}(\nabla F z_n).$$

Then,

$$z_2 = z_1 - (F''(x_1))^{-1}(\nabla F z_1) = \bar{0} - (F''(x_1))^{-1}(\nabla F(\vec{0})),$$

where we take $z_1 = \bar{0}$.

Since

$$\nabla F(\bar{z}) = F'(x_1) + F''(x_1)(\bar{z}),$$
$$\nabla F(\bar{0}) = F'(x_1) + F''(x_1)(\bar{0}),$$
$$\nabla F(\bar{0}) = F'(x_1).$$

Then,

$$z_2 = \bar{0} - (F''(x_1))^{-1} F'(x_1).$$

For convenience in exposition, we include formulas below from Hestenes [10], pp. 136–137 and pp. 199–202.

Here, the first step of Newton's method is applied to $\nabla F(\vec{z}) = \vec{0}$ and $z_2$ also turns out to be the only *min* of $F(z)$ (a quadratic equation with positive definite symmetric term), i.e.,

$$z_2 = -(F''(x_1))^{-1} F'(x_1),$$

which satisfies $\nabla F(z_2) = \vec{0}$. Therefore, Newton's method terminates in one iteration [10].

The initial formulas for $b_k$ and $c_k$ given in Algorithm 1 imply the basic CG relations

$$p_k^* r_{k+1} = 0, \qquad s_k^* p_{k+1} = 0.$$

---

**Algorithm 1** CG algorithm

---

Step 1: Select an initial point $x_1$. Set $r_1 = -f'(x_1)$, $p_1 = r_1$, $z_1 = 0$.
**for** $k = 1, \ldots, n$ **do** perform the following iteration:
   Step 2: $s_k = f''(x_1) p_k$,
   Step 3: $a_k = \frac{c_k}{d_k}$ , $d_k = p_k^* s_k$, $c_k = p_k^* r_k$ or $c_k = p_k^* r_1$,
   Step 4: $z_{k+1} = z_k + a_k p_k$, $r_{k+1} = r_k - a_k s_k$,
   Step 5: $p_{k+1} = r_{k+1} + b_k p_k$, $\qquad b_k = -\dfrac{s_k^* r_{k+1}}{d_k}$ or $b_k = \dfrac{|r_{k+1}|^2}{|r_k^2|}$.

**end for**
Step 6: **When** $k = n$ consider the next estimate of the minimum point $x_0$ of $f$ to be the point $\bar{x}_1 = x_1 + z_{n+1}$.
**Then** choose $\bar{x}_1$ as the final estimate, if $|f'(\bar{x}_1)|$ is sufficiently small enough.
**Otherwise**, reset $x_1 = \bar{x}_1$ and the CG cycle $(Step1)$–$(Step5)$ is repeated.

---

The CG cycle in Step 1 can terminate prematurely at the $m$th step if $r_{m+1} = 0$. In this case, we replace $x_1$ by $\bar{x}_1 = x_1 + z_{m+1}$ and restart the algorithm.

If we take $A = f''(x_1)$, where $A$ is positive definite symmetric, then we establish the formula

$$f''(x_1)^{-1} = \sum_{k=1}^{n} \frac{p_k p_k *}{d_k},$$

for the inverse of $f''(x_1)$.

Since Step 2 implies that $s_k = f''(x_1) p_k$, then, in Algorithm 1, we find

$$\lim_{\sigma \to 0} \frac{f'(x_1 + \sigma p_k) - f'(x_1)}{\sigma} = f''(x_1) p_k.$$

We obtain the difference quotient by rewriting the vector $s_k$ in Algorithm 1 (see Hestenes [10]). Therefore, without computing the second derivative we find

$$s_k = \frac{f'(x_1 + \sigma p_k) - f'(x_1)}{\sigma}.$$

In view of the development of Algorithms 1 and 2, each cycle of $n$ steps is clearly comparable to one Newton step.

Thus, we replace $c_k = p_k^* r_k$ by $c_k = p_k^* r_1$ and obtain the following relation

$$z_{n+1} = \sum_{k=1}^{n} \frac{c_k p_k}{d_k} = \sum_{k=1}^{n} \frac{p_k p_k^* r_1}{d_k} = H(x_1, \sigma)(-r_1) = -H(x_1, \sigma) f'(x_1),$$

where

$$H(x_1, \sigma) = \sum_{k=1}^{n} \frac{p_k p_k^*}{d_k}, r_1 = -f'(x_1).$$

---

**Algorithm 2** CG algorithm without derivative

---

Step 1: Initially select $x_1$ and choose a positive constant $\sigma$. Set $z_1 = 0$, $r_1 = -f'(x_1)$, $p_1 = r_1$.
**for** $k = 1, \ldots, n$ **do** perform the following iteration:

Step 2: $s_k = \dfrac{f'(x_1 + \sigma p_k) - f'(x_1)}{\sigma}, \quad \sigma_k = \dfrac{\sigma}{|p_k|},$

Step 3: $a_k = \dfrac{c_k}{d_k}, \quad d_k = p_k^* s_k, \quad c_k = p_k^* r_k,$

Step 4: $z_{k+1} = z_k + a_k p_k, \quad r_{k+1} = r_k - a_k s_k,$

Step 5: $p_{k+1} = z_k + a_k p_k, \quad b_k = -\dfrac{s_k^* r_{k+1}}{d_k}.$

**end for**
Step 6: **When** $k = n$, then $\bar{x}_1 = x_1 + z_{n+1}$ is to be the next estimate of the minimum point $x_0$ of $f$.
**Then** accept $\bar{x}_1$ as the final estimate of $x_0$, if $|f'(\bar{x}_1)|$ is sufficiently small.
**Otherwise**, reset $x_1 = \bar{x}_1$ and repeat the CG cycle (*Step*1)–(*Step*5).

---

The new initial point $\bar{x}_1 = x_1 + z_{n+1}$ generated by one cycle of the modified Algorithm 2 is, therefore, given by the Newton-type formula

$$\bar{x}_1 = x_1 - H(x_1, \sigma) f'(x_1).$$

So, we have $\lim_{\sigma \to 0} H(x_1, \sigma) = f''(x_1)^{-1}$. The above algorithm approximates the Newton algorithm

$$\bar{x}_1 = x_1 - f''(x_1)^{-1} f'(x_1)$$

and has this algorithm as a limit as $\sigma \to 0$. Therefore, Algorithm 2 will have nearly identical convergence features to Newton's algorithm if $\sigma$ is replaced by $\dfrac{\sigma}{2}$ at the end of each cycle.

*3.2. Conjugate Gram–Schmidt (CGS)—Algorithms for Nonquadratic Functions*

With an appropriate initial point $x_1$, we can derive the algorithm that is described by Hestenes [10] on p. 135, which relates Newton's method to a CGS algorithm. Since [10]

$$\lim_{\sigma \to 0} \frac{f'(x_1 + \sigma p_k) - f'(x_1)}{\sigma} = f''(x_1) p_k. \tag{11}$$

We can approximate the vector $f''(x_1) p_k$ by the vector

$$s_k = \frac{f'(x_1 + \sigma p_k) - f'(x_1)}{\sigma}, \tag{12}$$

with a small value of $\sigma_k$. Then, we obtain the following modification of Newton's algorithm, the CGS algorithm (see Hestenes [10]):
In Step 2 of Algorithm 3, substitute $s_k$ with the following formula

$$s_k = f''(x_1) p_k$$

and repeat the CGS algorithm. Then, we obtain Newton's algorithm.

---

**Algorithm 3** CGS algorithm

---

Step 1: Select a point $x_1$. a small positive constant, $\sigma > 0$ and $n$ linearly independent vectors $u_1, \ldots, u_n$; set $z_1 = 0$, $r_1 = -f'(x_1)$, $p_1 = u_1$.

**for** $k = 1, \ldots, n$ and having obtained $z_k$, $r_k$ and $p_k$ **do** perform the following iteration:

Step 2: $s_k = \dfrac{f'(x_1 + \sigma p_k) - f'(x_1)}{\sigma}$, $\sigma_k = \dfrac{\sigma}{|p_k|}$,

Step 3: $d_k = p_k^* s_k$, $c_k = p_k^* r_1$, $a_k = \dfrac{c_k}{d_k}$,

Step 4: $z_{k+1} = z_k + a_k p_k$,

Step 5: $b_{k+1, j} = \dfrac{s_j^* u_{k+1}}{d_j}$ $\qquad (j = 1, \ldots, k)$,

Step 6: $p_{k+1} = u_{k+1} - b_{k+1, 1} \, p_1 - \ldots - b_{k+1, k} \, p_k$.

**end for**

Step 7: **When** when $z_{n+1}$ has been computed, the cycle is terminated.

**Then** choose $\bar{x}_1$ as the final estimate, if $|f'(\bar{x}_1)|$ is sufficiently small enough.

**Otherwise**, reset $x_1 = \bar{x}_1$ and repeat the CGS cycle $(Step1)$–$(Step6)$.

---

In view of (11), for small $\sigma > 0$, the CGS Algorithm 3 is a good approximation of Newton's algorithm as a limit as $\sigma \to 0$.

A simple modification of Algorithm 3 is obtained by replacing the following formulas in Step 2 and Step 3, as described in Hestenes [10].

$$s_k = \frac{f'(x_1 + \sigma p_k) - f'(x_1)}{\sigma}, \sigma_k = \frac{\sigma}{|p_k|},$$

$$x_k = x_1 + z_k, d_k = p_k^* s_k, c_k = -p_k^* f'(x_k), a_k = \frac{c_k}{d_k}.$$

A CGS algorihtm for nonquadratic functions is obtained form the following relation, where the ratios

$$c(\sigma) = \frac{f(x - \sigma p) - f(x + \sigma p)}{2\sigma},$$

$$d(\sigma) = \frac{(f - \sigma p) - 2f(x) + f(x + \sigma p)}{\sigma^2},$$

have the properties

$$\lim_{\sigma \to 0} c(\sigma) = -p^* f'(x), \qquad \lim_{\sigma \to 0} d(\sigma) = p^* f''(x) p,$$

and $p$ is a nonzero vector. Moreover, for a given vector $u \neq 0$, the ratio

$$c(\alpha, \sigma) = \frac{f(x + \alpha u - \sigma p) - f(x + \alpha u + \sigma p)}{2\sigma},$$

has the property that

$$\lim_{\alpha \to 0} \lim_{\sigma \to 0} \frac{c(\sigma) - c(\alpha, \sigma)}{\alpha} = u^* f''(x) p.$$

The details are as follows. Suppose $p_1, p_2, \ldots, p_n$ is an orthogonal basis that spans the same vector space as that spanned by $u_1, u_2, \ldots, u_n$, which are linearly independent vectors. The inner product $(x, y)$ is defined by $x^* A y$, where $A$ is a positive definite symmetric matrix. Then, the Gram–Schmidt process works as follows:

$$\bar{p}_1 = u_1, \qquad\qquad p_1 = \frac{\bar{p}_1}{|\bar{p}_1|} = u_1$$

$$\bar{p}_2 = u_2 - \frac{(u_2, p_1)}{(p_1, p_1)} p_1, \qquad\qquad p_2 = \frac{\bar{p}_2}{|\bar{p}_2|}$$

$$\bar{p}_3 = u_3 - \frac{(u_3, p_1)}{(p_1, p_1)} p_1 - \frac{(u_3, p_2)}{(p_2, p_2)} p_2, \qquad\qquad p_3 = \frac{\bar{p}_3}{|\bar{p}_3|}$$

$$\bar{p}_3 = u_3 - \frac{(p_1^* A u_3)}{(p_1^* A p_1)} p_1 - \frac{(p_2^* A u_3)}{(p_2^* A p_2)} p_2,$$

$$\cdots$$

$$\bar{p}_{k+1} = u_{k+1} - \frac{(p_1^* A u_{k+1})}{(p_1^* A p_1)} p_1 - \cdots - \frac{(p_k^* A u_{k+1})}{(p_k^* A p_k)} p_k, \quad p_{k+1} = \frac{\bar{p}_{k+1}}{|\bar{p}_{k+1}|}.$$

Take $A = f''(x_1)$, then

$$\bar{p}_{k+1} = u_{k+1} - \frac{(p_1^* f''(x_1) u_{k+1})}{(p_1^* f''(x_1) p_1)} p_1 - \cdots - \frac{(p_k^* f''(x_1) u_{k+1})}{(p_k^* f''(x_1) p_k)} p_k.$$

We already proved that

$$p^* A p = D \quad \text{or} \quad p^* f''(x_1) p = D.$$

Then,

$$\bar{p}_{k+1} = u_{k+1} - \frac{(p_1^* f''(x_1) u_{k+1})}{d_1} p_1 - \cdots - \frac{(p_k^* f''(x_1) u_{k+1})}{d_k} p_k.$$

We also know that

$$p_k^* f''(x_1) = s_k.$$

Therefore,

$$\bar{p}_{k+1} = u_{k+1} - \frac{s_1 u_{k+1}}{d_1} p_1 - \cdots - \frac{s_k u_{k+1}}{d_k} p_k,$$

$$p_{k+1} = u_{k+1} - b_{k+1,1} \, p_1, \ldots, b_{k+1,k} \, p_k,$$

$$\bar{p}_{k+1} = u_{k+1} - b_{k+1,1} \, p_1, \ldots, b_{k+1,k} \, p_k, \quad \text{since} \; p_{k+1} = \frac{\bar{p}_{k+1}}{|\bar{p}_{k+1}|}.$$

Now using function values only, a conjugate Gram–Schmidt process without derivatives is described by Hestenes [10] as follows, as the CGS routine without derivatives (Algorithm 4):

---

**Algorithm 4** CGS algorithm without derivatives

---

Step 1: select an initial point $x_1$, small $\sigma > 0$ and a set of unit vectors $u_1, \ldots, u_n$, which are linearly independent; set $z_1 = 0$, $p_1 = u_1$, $\alpha = 2\sigma$, $\gamma_0 = 0$; compute $f(x_1)$.

**for** $k = 1, \ldots, n$ and having obtained $z_k, p_1, \ldots, p_k$ and $\gamma_{k-1}$, **do** perform the following iteration:

Step 2: $d_k = \dfrac{f(x_1 - \sigma p_k) - 2f(x_1) + f(x_1 + \sigma p_k)}{\sigma^2}$,

Step 3: $dc_k = \dfrac{f(x_1 - \sigma p_k) - f(x_1 + \sigma p_k)}{2\sigma}$,

Step 4: $\gamma_k = max[\gamma_{k-1}, |c_k|]$,

Step 5: $a_k = \dfrac{c_k}{d_k}$, $\qquad z_{k+1} = z_k + a_k p_k$,

Step 6: $p_{k+1} = u_{k+1} - b_{k+1,1}\, p_1 - \ldots - b_{k+1,k}\, p_k$.

**end for**

Step 7: **When** $z_{n+1}$ has been computed, the cycle is terminated.

**Then** choose $\bar{x}_1$ as the final estimate, if $|f'(\bar{x}_1)|$ is sufficiently small, $\bar{x}_1$ is the minimum of $f$.

**Otherwise**, reset $x_1 = \bar{x}_1$ and repeat the CGS cycle $(Step1)$–$(Step6)$ with the initial condition $\gamma_0 = 0$.

---

In addition, the conjugate Gram–Schmidt method without derivatives is described by Dennemeyer and Mookini [26]. In this program, they used different notations from Hestenes' notations, but they provided the same procedure.

Initial step: select an initial point $x_1$, a small $\sigma > 0$ and a set of linearly independent vectors $u_1, \ldots, u_n$;

set $h_1 = 0$, $\qquad p_1 = u_1$, $\qquad \alpha = 2\sigma$, $\qquad \gamma_0 = 0$ and compute $f(x_1)$.

Iterative steps: given $x_1, p_1, \ldots, p_k, h_k$, compute

$$d_k = \frac{f(x_1 - \sigma p_k) - 2f(x_1) + f(x_1 + \sigma p_k)}{\sigma^2},$$

$$c_k = \frac{f(x_1 - \sigma p_k) - f(x_1 + \sigma p_k)}{2\sigma},$$

$$\gamma_k = \max[\gamma_{k-1}, |c_k|], \qquad a_k = \frac{c_k}{d_k}, \qquad h_{k+1} = h_k + a_k p_k;$$

for $j = 1, \ldots, k$ compute

$$c_{k+1,j} = \frac{f(x_1 + \alpha u_j - \sigma p_k) - f(x_1 + \alpha u_j + \sigma p_k)}{2\sigma},$$

$$a_{k+1,j} = \frac{c_{k+1,j}}{d_j}, \qquad b_{k+1,j} = \frac{a_{k+1,j} - a_j}{\alpha},$$

then,

$$p_{k+1} = u_{k+1} + \sum_{j=1}^{k} b_{k+1,j}\, p_j.$$

Terminate when $h_{n+1}$ is obtained, and set $x_{n+1} = x_1 + h_{n+1}$. If the value $\gamma_n$ is small enough, $x_{n+1}$ is the minimum point of $f$. Otherwise, set $x_1 = x_{n+1}$ and repeat the program.

The term $\gamma_n$ is used to terminate the algorithm because the gradient is not explicitly computed. Another termination method would be to test if $\max |a_j| < \epsilon$ is chosen beforehand. Both of these tests were used on the computer by Dennemeyer and Mookini [26] and the results were comparable.

## 4. Discussion

In this section, we present a computation to illustrate convergence rates, as well as the relationship between that computation and Newton's method. Two of the most

important concepts in the study of iterative processes are the following: (a) when the iterations converge; and (b) how fast the convergence is. We introduce the idea of rates of convergence, as described by Ortega and Rheinboldt [14].

*4.1. Rates of Convergence*

A precise formulation of the asymptotic rate of convergence of a sequence $x^k$ converging to $x^*$ is motivated by the fact that estimates of the form

$$||x^{k+1} - x^*|| \leq ||x^k - x^*||^p, \tag{13}$$

for all $k = 1, 2, \ldots$, often arise naturally in the study of certain iterative processes.

**Definition 1.** *Let $x^k$ be a sequence of points in $R^n$ that converges to a point $x^*$. Let $1 \leq p < \infty$. Ortega and Rheinboldt [14] define the quantities*

$$\mathcal{Q}_p\{x^k\} = \begin{cases} 0 & \text{if } x^k = x^* \text{ for all but finitely many } k, \\ \limsup_{k \to \infty} \frac{||x^{k+1} - x^*||}{||x^k - x^*||^p} & \text{if } x^k \neq x^* \text{ for all but finitely many } k, \\ +\infty & \text{otherwise,} \end{cases}$$

*and refer to them as quotient convergence factors, or Q-factors for short.*

**Definition 2.** *Let $C(\mathcal{I}, x^*)$ denote the set of all sequences having a limit of $x^*$ that are generated by an iterative process $\mathcal{I}$.*

$$\mathcal{Q}_p(\mathcal{I}, x^*) = \sup\{\mathcal{Q}_p\{x^k\} | \{x^k\} \in C(\mathcal{I}, x^*)\} \qquad 1 \leq p < +\infty,$$

*are the Q-factors of $\mathcal{I}$ at $x^*$ with respect to the norm in which the $\mathcal{Q}_p\{x^k\}$ are computed.*

Note that if $\mathcal{Q}_p\{x_k\} < +\infty$ for some $p$ where $1 \leq p < \infty$, then, for any $\epsilon > 0$, there is some positive integer $K$ such that (13) above holds for $C = \mathcal{Q}_p\{x_k\} + \epsilon$. If $0 < \mathcal{Q}_p\{x_k\} < \infty$, then we say that $x^k$ converges to $x^*$ with Q-order of convergence p, and if $\mathcal{Q}_p\{x_k\} = 0$, for some fixed $p$ satisfying $1 \leq p < \infty$, then we say that $x^k$ has superconvergence of Q-order $p$ to $x^*$. For example, if $0 < \mathcal{Q}_p\{x_k\} < +\infty$ when $p = 1$, then we also have $0 < C < 1$ in (13), we say that $\{x_n\}$ converges to $x^*$ linearly. In addition, if $\mathcal{Q}_p\{x_k\} = 0$ when $p = 1$, we say that $\{x_n\}$ converges to $x^*$ superlinearly.

**Definition 3.** *One other method of describing convergence rate involves the root convergence factors. See ([14]).*

$$\mathcal{R}_p(x_n) = \begin{cases} \limsup_{k \to \infty} ||x_n - x^*||^{1/n} & \text{if } \quad p = 1, \\ \limsup_{k \to \infty} ||x_n - x^*||^{1/p^n} & \text{if } \quad p > 1. \end{cases}$$

*4.2. Acceleration*

One acceleration procedure is the following: first, apply $n$ CD steps to an initial point $x_1$ to obtain a point $x_{n+1} = y_1$; then, take $x_{n+1}$ to be a new initial point and apply $n$ CD steps again to obtain another $x_{n+1} = y_2$; finally, check for acceleration by evaluating $Q = F(y_2 - (Y_2 - y_1))$, if $Q < F(y_2)$; then, we accelerate by taking $[y_2 - (y_2 - y_1)]$ as our initial point; if $Q > F(y_2)$, then take $y_2$ as a new initial point; after two more applications of the CD method, we check for acceleration again. The procedure continues in this manner [25].

*4.3. Test Function*

4.3.1. Rosenbrook's Banana Valley Function

We carry out the following computations for Rosenbrook's banana valley function ($n = 2$). This function possesses a steep sided valley that is nearly parabolic in shape. First, we determine values in the domain of Rosenbrock's function for which its Hessian matrix is positive definite symmetric. Since the Rosenbrock's banana valley function is non-negative, i.e.,

$$f(x,y) = [100(y - x^2)^2 + (x - 1)^2] \geq 0,$$

then we have

$$f_x = 200(y - x^2)(-2x) + 2(x - 1) = -400x(y - x^2) + 2(x - 1),$$

and

$$f_{xx} = -400(y - x^2) - 400x(-2x) + 2 = -400y + 400x^2 + 800x^2 + 2 = 1200x^2 - 400y + 2,$$

and

$$f_{xy} = -400x, \qquad f_y = 200(y - x^2), \qquad f_{yy} = 200.$$

Therefore, the Hessian matrix is positive definite symmetric if and only if Sylvester's criterion holds:

$$(1200x^2 - 400y + 2) > 0, \quad \text{and} \quad \left((200)(1200x^2 - 400y + 2) - 160000x^2\right) > 0,$$

which implies that $1200x^2 + 2 > 400y, \Leftrightarrow y < 3x^2 + \frac{1}{200}$, and

$$1200x^2 - 400y + 2 - 800x^2 > 0, \Leftrightarrow 400x^2 + 2 > 400y, \Leftrightarrow y < x^2 + \frac{1}{200}.$$

So, the Hessian matric is positive definite symmetric if and only if $y < x^2 + \frac{1}{200}$.

Figure 1 shows the maximal convex level set on which the Hessian is positive definite symmetric in the interior for Rosenbrock's Banana Valley Function.

4.3.2. Kantorovich's Function

The following function

$$F(x,y) = (3x^2y + y^2 - 1)^2 + (x^4 + xy^3 - 1)^2,$$

which is non-negative, i.e., $F(x_1, x_2) \geq 0$, is called Kantorovich's Function.

Calculating the Hessian matrix for Kantorovich's function, we find that

$$F_{xx} = 72x^2y^2 + 12(3x^2y + y^2 - 1)y + 2(4x^3 + y^3)^2 + 24(x^4 + xy^3 - 1)x^2,$$

$$F_{xy} = 12(3x^2 + 2y)xy + 12(3x^2y + y^2 - 1)x + 6xy^2(4x^3 + y^3) + 6(x^4 + xy^3 - 1)y^2$$

and

$$F_{yy} = 2(3x^2 + 2y)^2 + 12x^2y + 4y^2 - 4 + 18x^2y^4 + 12(x^4 + xy^3 - 1)xy.$$

Minimizing this function is equivalent to solving the nonlinear system of equations. Therefore, for the initial point $(0.98, 0.32)$, we obtain the minimum point at $(0.992779, 0.306440)$ [25].

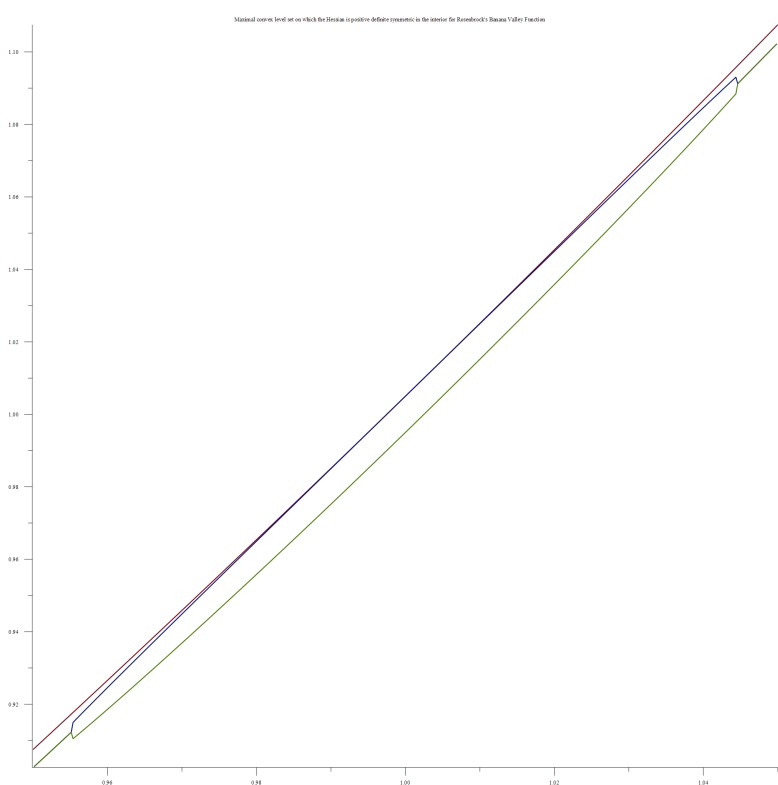

**Figure 1.** Maximal conves level set for Rosenbrock's banana valley function.

### 4.4. Numerical Computation

The goal of this numerical computation is to provide a system of iterative approaches for finding these extreme points [10]. A significant point is that a Newton step can be performed instead by a CD sequence of $n$ linear minimizations in $n$ appropriately chosen directions.

It is important to keep in mind that a function acts like a quadratic function when it is in the neighborhood of a nondegenerate minimum point. Conjugacy can be thought of as a generalization of the concept of orthogonality. Conjugate direction methods include substituting conjugate bases for orthogonal bases in the foundational structure. The formulas for determining the minimum point of a quadratic function can be reduced to their simplest forms by following the CD technique.

The conjugate direction algorithms for minimizing a quadratic function, which are discussed in the current work, were initially presented in Hestenes and Stiefel, 1952 [5]. These algorithms can be found in the present work. The authors Davidon [3], Fletcher and Powell [4] are most known for the modifications and additions that they made to these methods. However, numerous other authors also made these changes.

The iterative methods described above apply to many problems. They are used in least squares fitting, in solving linear and nonlinear systems of equations and in optimization problems with and without constraints [25]. The computing performances and numerical results of these techniques and comparisons have received a significant amount of attention in recent years. This interest has been focused on the solving of unconstrained optimization problems and large-scale applications [19,27].

The Rosenbrock function of two variables, considered in Section 4.3, was introduced by Rosenbrock [18] as a simple test function for minimization algorithms. We chose $(x_1, y_1) = (-1.2, 1)$ as the initial point. We applied algorithm $(4.4a)$–$(4.4f)$ with $\sigma = 0.1 \times 10^{-120}$, using 400-digit accuracy. Algorithm (4) is basically Newton's algorithm.

The final estimate of $(x_0, y_0)$ has more than 150-digit accuracy. The successive values $0.8574\ldots, 0.0274\ldots, 0.2433\ldots, 0.0030\ldots, 0.2000\ldots, 0.0030\ldots, 0.2000\ldots, \ldots$ of quotients

that lead to the quotient convergence factor oscillate. The lim sup of these quotients give the quotient converge factor, which indicates quadratic convergence. The lim sup is .2000 . . ..

For $\sigma = 0.1 \times 10^{-120}$, $\rho = 0.2 \times 10^{-120}$, $\epsilon = 0.1 \times 10^{-60}$ and the initial values, we obtained the following computations for Rosenbrock's function $f$ using the Gram-Schmidt Conjugate Direction Method without Derivatives or the CGS method, no derivatives, and Newton's Method applied to $\nabla f = 0$: (See [28])

For additional information regarding the programming, please refer to the supplementary material.

*4.5. Differential Equations of Steepest Descent*

The following equations are known as the differential equations of steepest descent:

$$\frac{dx(t)}{dt} = -\nabla F(x(t)), \tag{14}$$

and

$$\frac{dx(t)}{dt} = \frac{-\nabla F(x(t))}{||\nabla F(x(t))||_2}. \tag{15}$$

The solution to either differential equation of steepest descent with initial condition $x_1(0) = -1.2$, $x_2(0) = 1.0$ is shown in Figure 2, one can refer to Equation (10), p. 783, in Eells [17]. For Equation (14), the solution will not include the minimum for finite values of $t$. For Equation (15), the solution will approach the minimum, but will blow up at the minimum.

From a numerical point of view, the differential equation approach has to be used with caution. Rosenbrock [15] pointed out that the iterative method of steepest descent with line searches was not effective with steep valleys. The iterative method was introduced by Cauchy [16].

In summary, the method of steepest descent is not effective and does not compare with Hestenes' CGS method with no derivatives, which is almost numerically equivalent to Newton's method applied to grad$(f) = 0$, where $f$ is the function to be minimized.

Below are level curves of Rosenbrock's banana valley function. We used this function to compare Hestenes' CGS method, Newton's method and the steepest descent methods. In Figure 2, the level curves of Rosenbrock's Banana Valley Function show that the minimizer is at $(1, 1)$. Level curves are plotted for function values $4.0, 4.1, 4.25, 4.5$ in Figure 3. For steepest descent, the iterative method and the ODE approach are illustrated. The curve $y = x^2$ appears to parallel the valley floor in the graph.

We use the CGS method for computation. The Rosenbrock's banana valley function

$$F(x_1, x_2) = (1 - x_1)^2 + 100(x_2 - x_1^2)^2,$$

gives the minimum point at $(1, 1)$.

This example provided us with geometric illustrations in Figure 2. For specific algorithms, please refer to Section 3 for the Gram–Schmidt conjugate direction method and the Newton method in order to compare the two methods along side one another.

The outcomes of the numerical experiments performed on the standard test function using the CGS method are reported above. Based on these data, it is clear that this particular implementation of the CGS method is quite effective.

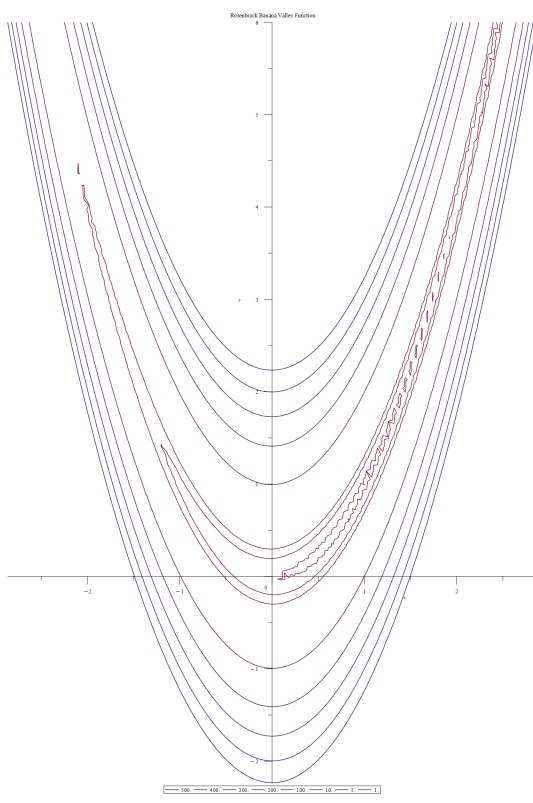

**Figure 2.** Level Curves of Rosenbrock's banana valley function.

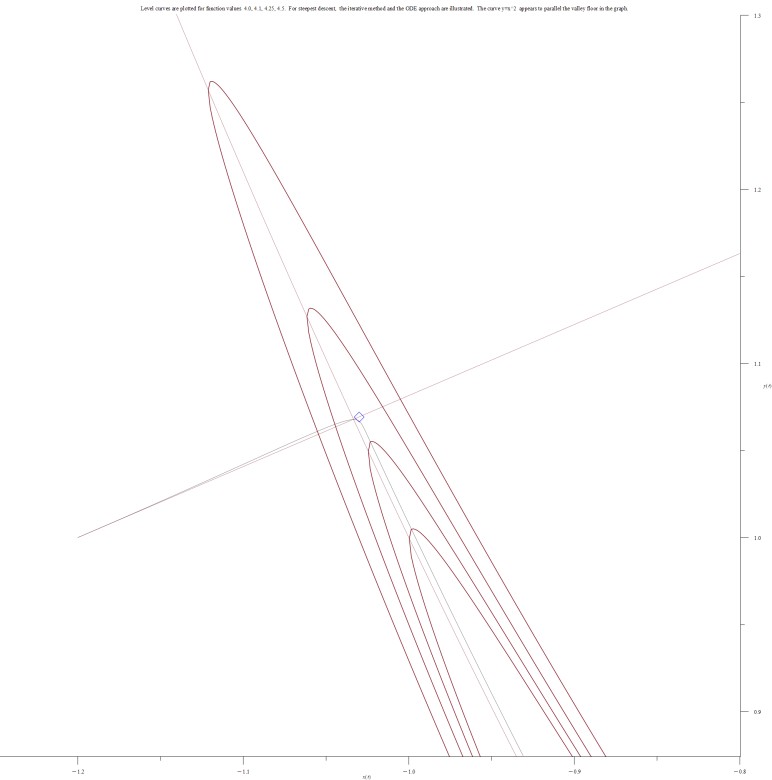

**Figure 3.** Curve of steepest descent and level curves for Rosenbrock's banana valley function.

## 5. Conclusions

In this paper, we introduced a class of CD algorithms that, for small values of $n$, provided effective minimization methods. As $n$ grew, however, the algorithms became more and more costly to run.

The computer program above showed that the CGS algorithm without derivatives could generate Newton's method. Since the Hessian matrix of Rosenbrock's function was positive definite symmetric and satisfied Sylvester's criterion, the CGS method converged if we began anywhere in the closed convex set in the nearby area of a minimum. This was because the CGS method is based on the fact that the Hessian matrix of Rosenbrock's function is positive definite symmetric.

Using quotient convergence factors, one can see that for Rosenbrock's function one sequence converged quadratically. In particular, the numerical computation on p. 21 revealed that the asymptotic constant oscillated between 0.20000 and 0.00307, so the quotient convergence factor by Ortega and Rheinboldt [14] was, approximately, $\mathcal{Q}_2\{x^k\} = 0.200002$, which indicated quadratic convergence. The results agreed for Newton's method.

Moreover, the CGS algorithm uses function evaluations and difference quotients for gradient and Hessian evaluations, it does not require accurate gradient evaluation nor function minimization. This approach is the most efficient algorithm that has been discussed in this study; yet, it is extremely sensitive to both the choice of $\sigma$ that is used for difference quotients and the choice of $\rho$ that is used for scaling.

The Gram–Schmidt conjugate direction method without derivatives has been used quite successfully in a variety of applications, including radar designs by Norman Olsen [27] in developing corporate feed systems for antennas and aperture distributions for antenna arrays. He tweaked the parameters sigma and rho in our GSCD computer programs to obtain successful radar designs.

**Supplementary Materials:** Supporting information can be downloaded at: https://www.mdpi.com/article/10.3390/appliedmath3020015/s1.

**Author Contributions:** Conceptualization, I.S.J. and M.N.R.; methodology, I.S.J.; software, I.S.J.; validation, M.N.R. and I.S.J.; formal analysis, M.N.R.; investigation, M.N.R.; resources, I.S.J.; data curation, I.S.J.; writing—original draft preparation, I.S.J.; writing—review and editing, M.N.R.; visualization, M.N.R.; supervision, I.S.J.; project administration, I.S.J. All authors have read and agreed to the published version of the manuscript.

**Funding:** This research received no external funding.

**Institutional Review Board Statement:** Not applicable.

**Informed Consent Statement:** Not applicable.

**Data Availability Statement:** Not applicable.

**Conflicts of Interest:** The authors declare no conflict of interest.

### Acronyms

| | |
|---|---|
| CD | conjugate direction; |
| CG | conjugate gradient; |
| CGS | conjugate Gram–Schmidt; |
| GSCD | Gram–Schmidt conjugate direction. |

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
