# Peer review of "Convergence Rates for Hestenes’ Gram–Schmidt Conjugate Direction Method without Derivatives in Numerical Optimization"

_2673-9909, doi:10.3390/appliedmath3020015_

Round 1

Reviewer 1 Report

see the review in attachment file.

Author Response

Dear Reviewer,

Thank you so much for taking time to reading our article and providing feedback on it. We really appreciate it.

Point 1: We have made the necessary adjustments to the introduction section after reviewing your comments.

Point 2: Both the author's name, Polyak, and the title have been updated.

Point 3: We have supplemented the review of the study topic with publications with recent publications during the previous 15 years. However, we have tried to locate anybody as much as who has done this over the past 5 years. 

Please, let us know, if there is anything we can improve.

Best regards,

Nurul Raihen, Ivie Stein

Reviewer 2 Report

This paper presents an interesting overview of convergence rates using quotient and root convergence factors for Hestenes' Gram-Schmidt conjugate direction method without derivatives. The authors perform a computational and analytical comparison between the different methods and provide important additional details and computer code. The paper is clear and concise and can prove valuable for the community.

In my view, the calculations and derivations seem correct, although I did not check them in full detail since they are based on well-established references such as [8]. I think this is a worthwhile contribution, and I recommend that the paper be published in Applied Math.

Following, I include some minor comments for the authors:

- The word "nonquadratic" appears on two occasions (p. 1 and 2) as opposed to "non quadratic" in the rest of the occurrences. I, however, suggest the hyphenated form "non-quadratic".

- The figure on page 19 is taken from reference [8] (authored by the first author) without acknowledgment. I recommend adding a caption to the figure mentioning it is taken from [8]. 

- Reference [8] is not in alphabetical order with the rest.

Author Response

Dear Reviewer,

Thank you so much for taking time to read our article and providing feedback on it. We really appreciate it.

Point1: We have fixed the word from nonquadratic to non-quadratic.

Point2: We have updated the figure to include a caption and recognized Ivie Stein as the author who developed it.

Point3: Orders are being placed for all References.

Please, let us know if you have any questions.

Best regards,

Nurul Raihen, Ivie Stein

Reviewer 3 Report

Comments and suggestions

-The number of references is few. Please add more references.

- A quote from recent sources for the years 2021-2022.

-Solve more unrestricted optimization test functions to demonstrate the efficiency of the proposed method.

-Figure quality is very poor.

- There is no title for the Figure.

- Make a table of comparisons of the numerical results with other methods in the same field.

Best Regards,

Author Response

Dear Reviewer,

Thank you so much for taking time to reading our article and providing feedback on it. We really appreciate it.

Point1. We add more references.

Point2. We add quotes from Andrei, Nedulai, June 2021, and other new sources.

Point3. Using the CGS technique, we add additional test functions.

Point4. The overall quality of our figures has increased.

Point5. The figure now has a title that we've added.

Point6. Along with Professor Stein, I demonstrated, using both computational and analytical methods, that the CGS technique converges in a quadratic fashion in comparison to Newton's Method. The article has been updated to include our program with explanation.

Thank you again. Please, let us know if you have any questions. 

Best regards,

Nurul raihen, Ivie Stein

Reviewer 4 Report

My comments are as follows:

1. In the Keyword: just maintain the Conjugate direction method or Conjugate gradient method, one of the two is enough. Also, the term convergence rates should be added.

2. In the introduction lines 16-17, a citation is needed for the statement “Variable metric methods are considered by many to be the most effective technique for optimizing a nonquadratic function.”

3. I recommend that the abbreviations of CG for Conjugate gradient and CD for conjugate direction should be used throughout the article after the first introduction or you can decide to ignore the abbreviations throughout the article.

4. Since this study is highly connected with the convergence rates using quotient convergence factors and root convergence factors, I expect the introduction section to talk about different convergence rates for CG, CD, and other related algorithms in the introduction section.

5. All equations should be fully punctuated, e.g. by a comma (,) or a full stop (.).

6. Some recently proposed derivative-free methods can be discussed to strengthen the introduction sections such as:

https://doi.org/10.1016/j.apnum.2022.10.016

7. The result discussion should be elaborated in detail to express the advantages of this work over the existing ones.

I recommended a revision.

Author Response

Dear Reviewer,

Thank you so much for taking time to read our article and providing comments on it. We really appreciate it.

Point1. We have adjusted the key word. 

Point2. Lines 16-17 have been expanded by Citation.

Point3. We have adjusted CG and CD after the introduction section. 

Point4. In the introduction section, we have included explanations of different types of convergence rates.

Point5. All equations are punctuated now. 

Point6. Recent works and references have been added to the introduction.

Point7. The discussion of the results has been expanded upon.

Thank you. Please, let us know if you have any questions.

Best regards,

Nurul Raihen, Ivie Stein

Reviewer 5 Report

1. The organization of the manuscript needs serious revision. This will aid the understanding of the potential readers.

2. Many equations are not properly punctuated, see, for example, eq. (2.1), (2.2), (2.3)... etc

3. The motivation of the manuscript should be clearly stated at the end of the first Section.

4. More explanation should be provided in Subsection 4.4.

5.  More recent relevant references should be added.

Author Response

Dear Reviewer,

Thank you so much for taking time to read our article and providing comments on it. We really appreciate it.

Point1. To facilitate reading of the article, the framework has been set up accordingly.

Point2. Equations have the appropriate punctuation throughout.

Point3. The motivation, as stated at the end of the first section is that Newton's method has the desirable feature of rapid convergence but has the undesirable feature that each step requires a significant amount of computing, particularly when matrix inversion is used explicitly. The formulas for determining the minimum point of a quadratic function can be reduced to their simplest forms by the following CGSDM technique and use this technique we show that computationally the function converges quadratically in compare with the Newton's Method.

Point4. Subsection 4.4 now includes more clarification.

Point5. The list of references has been updated to include more recent and pertinent works.

Thank you. Please let's know if you have any questions. 

Best regards,

Nurul Raihen, Ivie Stein